Remarkable dominance of myctophid otoliths in Upper Miocene Chagres Formation, Caribbean Panama

Lin Chien-Hsiang chlin.otolith@gmail.com 1
O’Dea Aaron 2 3
1 Biodiversity Research Center, Academia Sinica , Taipei , Taiwan
2 Smithsonian Tropical Research Institute , Panama City , Panama
3 Sistema Nacional de Investigación , Panama City , Panama
De Baets Kenneth
Electronic publication date: 2025 Oct 17
Publication date: 2025
Volume: 13
Electronic Location ID: e20155
Received 2025 May 21; Accepted 2025 Sep 8
Copyright: ©2025 Lin and O’Dea
Copyright year: 2025
Copyright holder: Lin and O’Dea
License: This is an open access article distributed under the terms of the Creative Commons Attribution License, which permits unrestricted use, distribution, reproduction and adaptation in any medium and for any purpose provided that it is properly attributed. For attribution, the original author(s), title, publication source (PeerJ) and either DOI or URL of the article must be cited.
License URL: https://creativecommons.org/licenses/by/4.0/

Keywords: Chagres formation, Neogene, Mesopelagic fish, Trophic transfer, Diversity, Upwelling, New species

Funding: National Science and Technology Council, Taiwan Grant No. 112-2116-M-001-017-MY3 Academia Sinica, Taipei, Taiwan AS-CDA-114-L04 Smithsonian Tropical Research Institute and the National Science Foundation EAR-2347773 This research was supported by a travel grant from the National Science and Technology Council, Taiwan (Grant No. 112-2116-M-001-017-MY3), and by Academia Sinica, Taipei, Taiwan (Career Development Award, AS-CDA-114-L04) to Chien-Hsiang Lin. This work was supported by grants from the Smithsonian Tropical Research Institute and the National Science Foundation (EAR-2347773) to Aaron O’Dea. The funders had no role in study design, data collection and analysis, decision to publish, or preparation of the manuscript.

==============================
Marine fossils from the Upper Miocene Chagres Formation in northern Panama offer critical insights into the paleoenvironmental conditions and ecological responses prior to the separation of the Atlantic from the Pacific by the formation of the Isthmus of Panama. Here we present a systematic study based on more than 6,200 otoliths collected from a coastal exposure near the town of Piña, Colón. This assemblage is remarkable for the extraordinary dominance of the family Myctophidae, constituting over 96% of specimens. The otolith density in the sediments is among the richest known globally (278.80 ± 135.59 otoliths/kg). The taxonomic composition is represented by 31 taxa across 12 families, including four new species: namely Chiloconger aflorens sp. nov., Dasyscopelus inopinatus sp. nov., Hoplostethus boyae sp. nov., and Malakichthys schwarzhansi sp. nov. Taphonomic evidence, combined with abundant predatory marine vertebrate fossils and extensive burrow ichnofossils, indicates a dynamic and highly productive nearshore ecosystem. The dominance of myctophids and multiple lines of evidence support the existence of a Late Miocene coastal upwelling system in the region, highlighted by efficient trophic transfer channeled from high primary production to apex predators. These findings provide a nuanced understanding of Neogene marine ecosystems prior to the final emergence of the Isthmus of Panama.

Introduction

The formation of the Isthmus of Panama is recognized as a critical event that fundamentally shaped modern ocean circulation, biogeographic patterns, and the evolutionary history of both terrestrial and marine organisms (Haug & Tiedemann, 1998; O’Dea et al., 2016; Stigall et al., 2017; Domingo et al., 2020; Jackson & O’Dea, 2023; Titus et al., 2024). Localized tectonic fracturing, faulting and extension, as well as subduction-driven uplift during the formation of the isthmus resulted in numerous marine sedimentary basins to develop initially as interconnected fore- and back-arc basins along an extended archipelago (Farris et al., 2011). Continued uplift and volcanic “infilling” (Buchs et al., 2019) eventually led to the isthmus forming a complete marine barrier in the Late Pliocene around 2.8 Ma (O’Dea et al., 2016). These sedimentary basins are crucial repositories of Neogene fossils, which have been helpful in revealing patterns of change in the diversity, ecology and evolution of marine faunas through a major environmental transition (Jackson & O’Dea, 2023).

Marine sedimentary archives often contain fish otoliths which offer unique insights into the spatiotemporal distribution, community structure, and evolutionary history of fishes (Nolf, 2013). In Tropical America, multiple studies have been conducted on Neogene otolith, including Schubert (1908), Gillette (1984), and Martin & Dunn (2000) in Panama, Nolf (1976) in Trinidad, Nolf & Stringer (1992) in the Dominican Republic, Stringer (1998) in Jamaica, Nolf & Aguilera (1998) and Aguilera & Rodrigues de Aguilera (2001), Aguilera & Rodrigues de Aguilera (2003) in Venezuela, and Aguilera et al. (2014) in Brazil. Studies by Schwarzhans & Aguilera (2013), Schwarzhans & Aguilera (2016), Schwarzhans & Aguilera (2024) and Aguilera, Schwarzhans & Béarex (2016), Aguilera et al. (2020) have comprehensively described otoliths of multiple families from tropical America, including material from Panama. Nonetheless, the Upper Miocene Chagres Formation, one of the significant fossil-yielding strata associated with the final stages of isthmus formation (Stiles et al., 2022), has not been investigated in its entirety with respect to its otolith assemblage at the community level.

The depositional environment of Chagres Formation was originally interpreted as a deep-water setting with notable Pacific influence (Collins & Coates, 1993; Coates & Obando, 1996), based primarily on the composition of benthic foraminifera (Collins et al., 1996; Collins et al., 1999), elasmobranch teeth (Carrillo-Briceño et al., 2015), and teleost otoliths (Aguilera & Rodrigues de Aguilera, 1999). Conversely, a recent study of ichnofossils and sedimentological data has proposed a relatively shallow-water depositional environment (Stiles et al., 2022).

In this study, we present results from quantitative otolith sampling from the Chagres Formation on the Caribbean coast of Panama. Based on the collection of over 6,200 otoliths, we document the richest fish records from the Late Miocene of the Panama Canal Basin. We provide detailed taxonomic remarks and assess taphonomic conditions which bring greater insights into the paleoenvironmental conditions and marine trophic dynamics of the region during the Late Miocene.

Geological setting

The Panama Canal Basin is underlain primarily by Cretaceous volcanic and plutonic rocks, which reflects its volcanic arc origins (Coates & Obando, 1996; Coates, 1999). Overlying this basement is a series of Cenozoic sedimentary formations, and the extensive Neogene transisthmian marine sediments (i.e., deposits spanning across the Isthmus of Panama) were exposed during the construction of the Panama Canal (Woodring, 1957). Sediment deposition within the basin is structurally controlled by the Gatun Fault Zone, which separates the Chorotega Block in western Panama and the Choco Block in the east (Coates & Obando, 1996; Coates, 1999).

On the Caribbean side of Colón Province, Panama, the Neogene deposits are primarily represented by the Gatun Formation and the overlain Chagres Formation (Coates, 1999). The Gatun Formation is of late Middle to Late Miocene (Hendy, 2013), consists of approximately 500 m of massive, blue-gray, marine fine sandstone to siltstone, notable for its rich and diverse marine fossils, particularly the mollusk fossils (Woodring, 1957; Jackson et al., 1999). Fossil evidence indicates deposition occurred at relatively shallow marine depths, typically less than 40 m (Coates & Obando, 1996; Collins et al., 1999).

The Chagres Formation, deposited between approximately 6.4 and 5.8 Ma (Late Miocene; Collins et al., 1996), is predominantly exposed in the northern part of the Panama Canal Zone and extends southwestward along the Caribbean coast (Fig. 1). The formation is about 250 m thick and is primarily comprised of marine, blue-gray volcaniclastic sandstone derived from volcanic arcs (Collins et al., 1996; Coates, 1999). It is subdivided into three members: the lower Toro Limestone, the middle Rio Indio Siltstone, and the upper Chagres Sandstone members. The Toro Limestone Member is characterized by calcareous beds rich in coquina composed of echinoid, mollusk, and barnacle fragments (Coates, 1999; Stiles et al., 2022). The Rio Indio Siltstone is characterized by gray-brownish silts with scattered mollusk fossils (Stiles et al., 2022). The Chagres Sandstone Member consists of gray, quartzose, volcanic-derived silty sandstones that exhibit extensive bioturbation, predominantly from arthropod burrows (Collins et al., 1996; Stiles et al., 2022). These sandstones yield abundant marine fossils and are prominently exposed along the Caribbean coast (Aguilera & Rodrigues de Aguilera, 1999; Carrillo-Briceño et al., 2015; Pyenson et al., 2015; Velez-Juarbe et al., 2015; Stiles et al., 2022; Benites-Palomino et al., 2023; Cadena, Gracia & Combita-Romero, 2023; Aguilera et al., 2025), particularly near the town of Piña (see below).

Figure 1 Sampling site and geological map of the Caribbean coastal region on the Isthmus of Panamá around the Piña site showing extent of Chagres Formation.

Figure modified after Collins et al. (1996) and Carrillo-Briceño et al. (2015).

Materials & Methods

Sampling

The fossil site is located about ∼500 m northeast of Piña town, Colón, Panama (Fig. 1; 9°17′09.11″N, 80°02′41.28″W). This locality corresponds to the Piña Norte site described by Stiles et al. (2022) and the STRI site 650009 in Pyenson et al. (2015). The Chagres Sandstone Member of the Chagres Formation is exposed along the Caribbean shoreline, especially at low tide (Fig. 2B). Fragments of echinoids, mollusks, and fish otoliths, as well as trace fossils, are easily visible on the surface of the siltstone (Stiles et al., 2022). Surface sediments are yellowish to brownish, contrasting with fresh sediments located several centimeters below the surface, which are darker and have a distinct blue-gray color. We collected thirty-three bulk sediment samples laterally from the same stratigraphic level in this fresher, blue-grey material in 2018 and 2024 (Fig. 2A). Samples weighed on average 0.6 kg each (Table S1). An additional larger bulk sample (unweighted but less than 2 kg, CH18-1-1) was also collected from this fresh sandstone layer for otolith extraction (Tables S1, S2). Permits for collecting and exporting paleontological samples were issued by the Ministerio de Comercio e Industrias (MICI, SE/AO-4-18) in Panamá.

Figure 2 Stratigraphic section and observed fossils in the Piña site of the Chagres Formation (A, modified after Carrillo-Briceño et al., 2015; Stiles et al., 2022) and photographs of the site (B–D) (C and D) Abundant fish otoliths and associated ichnofossil Ophiomorpha are visible on the surface of the outcrop.

Red arrow = sampling layer.

Otolith preparation, imaging, and identification

Bulk sediment samples were disaggregated using freeze-thaw cycles and Glauber’s Salt (saturated sodium sulfate solution) methods (Hanna & Church, 1928; Herrig, 1966), then wet-sieved through a 500-µm mesh. After sieving, sediments were dried overnight in an oven at 40 °C. Otoliths larger than this 500-µm mesh were carefully hand-picked under a stereomicroscope. In this study, the term “otolith” refers to the saccular otolith (sagitta). Representative otoliths were photographed using a digital camera adapted to a Nikon SMZ1270 stereomicroscope, and image stacking was performed using Helicon Focus software. Final figures were prepared using Adobe Photoshop.

For otolith identification and terminology, we followed key references, including Rivaton & Bourret (1999), Lin & Chang (2012), Schwarzhans (2013b), Schwarzhans & Aguilera (2013); Schwarzhans & Aguilera (2016), Nolf (2013), and Haimovici et al. (2024). In addition, direct comparisons were made with a reference collection of extant otoliths housed at the Biodiversity Research Museum, Academia Sinica, Taiwan (BRCAS) under the code CHLOL. Whenever possible, otoliths were identified to the species level. All collected specimens are stored at BRCAS, and figured specimens are archived under the registration code ASIZF.

Sampling completeness and biodiversity analysis

Due to the overwhelming number of otoliths in each sample, we conducted biodiversity analysis to assess sampling completeness and diversity. Otolith abundances were quantified by dividing otolith counts by the corresponding dry sediment weight (kg) for each sample (except CH18-1-1). Family-level abundances were calculated by summing otolith counts across all samples, and families were ranked by total abundance. To evaluate statistical uncertainty in these abundance estimates, binomial 95% confidence intervals were computed using Wilson’s method. We computed these with and without the unweighted sample CH18-1-1. Diversity was estimated using Hill numbers (Hill, 1973) calculated at three different orders: q = 0 (0D, species richness), q = 1 (1D, Shannon diversity), and q = 2 (2D, Simpson diversity), representing total (alpha) species richness, abundant species diversity, and dominant species diversity, respectively (Chao, Chiu & Jost, 2014; Chao et al., 2020; Lin et al., 2023a). Rarefaction and extrapolation methods were applied to create species accumulation curves, with 1,000 bootstrap resampling iterations used to estimate 95% confidence intervals. Specimen-based abundance data was analyzed to evaluate sample coverage comprehensively. All analyses were conducted using the R package iNEXT (Chao, Chiu & Jost, 2014; Hsieh, Ma & Chao, 2016).

Nomenclatural acts for new species

The electronic version of this article in Portable Document Format (PDF) will represent a published work according to the International Commission on Zoological Nomenclature (ICZN), and hence the new names contained in the electronic version are effectively published under that Code from the electronic edition alone. This published work and the nomenclatural acts it contains have been registered in ZooBank, the online registration system for the ICZN. The ZooBank LSIDs (Life Science Identifiers) can be resolved and the associated information viewed through any standard web browser by appending the LSID to the prefix http://zoobank.org/. The LSID for this publication is: urn:lsid:zoobank.org:pub:996A25D1-9CB7-4AAD-9041-0ABCF49710C5. The online version of this work is archived and available from the following digital repositories: PeerJ, PubMed Central and CLOCKSS.

Results

Systematic paleontology

A list of identified taxa and their abundances is presented in Table 1. Classification scheme follows Nelson, Grande & Wilson (2016). Morphometrics and measurements include otolith length (OL), otolith height (OH), sulcus length (SuL), ostium length (OsL), and cauda length (CaL). Descriptions and discussions for common or previously described taxa are provided briefly under remarks, while new species include detailed descriptions and diagnostics.

Order Anguilliformes	
Family Congridae	
Genus Chiloconger Myers & Wade, 1941	
Chiloconger aflorens sp. nov.	
(Figs. 3A–3B)	

Holotype: ASIZF 0100943 (Fig. 3A), Piña Norte, Panama. Upper Miocene, Chagres Formation. OL = 5.48 mm, OH = 4.23 mm.

Table 1 List of otolith-based fish taxa from the Upper Miocene Chagres Formation, Caribbean Panama.

Family	Taxa	No. of otoliths	
Congridae	Chiloconger aflorens sp. nov.	2	
	Rhynchoconger sp.	1	
	Congridae indet.	1	
Argentinidae	Argentina sp.	1	
Sternoptychidae	Polyipnus sp.	97	
Myctophidae	Benthosema pluridens Schwarzhans & Aguliera, 2013	44	
	Bolinichthys sp.	2	
	Dasyscopelus degraciai (Schwarzhans & Aguliera, 2013)	256	
	Dasyscopelus inopinatus sp. nov.	63	
	Diaphus aequalis Schwarzhans & Aguliera, 2013	125	
	Diaphus apalus Schwarzhans & Aguliera, 2013	152	
	Diaphus barrigonensis Schwarzhans & Aguliera, 2013	39	
	Diaphus dumerilii (Bleeker, 1856)	119	
	Diaphus multiserratus Schwarzhans & Aguliera, 2013	24	
	Diaphus pedemontanus (Robba, 1970)	78	
	Diaphus rodriguezi Schwarzhans & Aguliera, 2013	782	
	Diogenichthys sp.	1	
	Lepidophanes inflectus Schwarzhans & Aguliera, 2013	9	
	Lobianchia johnfitchi Schwarzhans & Aguliera, 2013	62	
	Myctophum affine (Lütken, 1892)	13	
	Myctophum arcanum Schwarzhans & Aguliera, 2013	24	
	Myctophidae indet.	4,179	
Macrouridae	Coelorinchus sp.	2	
	Macrouridae indet.	1	
Bregmacerotidae	Bregmaceros sp.	96	
Trachichthyidae	Hoplostethus boyae sp. nov.	4	
Carapidae	Carapus sp.	5	
Ophidiidae	Lepophidium limulum Schwarzhans & Aguliera, 2016	2	
Opistognathidae	Opistognathus sp.	1	
Carangidae	Carangidae indet.	2	
Malakichthyidae	Malakichthys schwarzhansi sp. nov.	15	
indet.	indet.	9	
	total	6,211	

Figure 3 Otoliths of Congridae and Argentinidae from the Upper Miocene Chagres Formation, Caribbean Panama.

(A and B) Chiloconger aflorens sp. nov., (A) holotype, ASIZF 0100943, (B) paratype, ASIZF 0100944. (C) Rhynchoconger sp., ASIZF 0100945. (D) Argentina sp., ASIZF 0100946. Images are inner views unless otherwise indicated. 1, ventral view; 2, inner view. Scale bar = one mm.

Paratype: One specimen: ASIZF 0100944 (Fig. 3B), same data as holotype. OL = 1.85 mm, OH = 1.52 mm.

Etymology: The species name aflorens is derived from the Spanish word “afloramiento”, meaning “upwelling”. It refers to the flourishing productivity and dynamic marine conditions of the coastal upwelling system in which this species lived. It also symbolically reflects the scientific blossoming of paleontological research in Panama.

Diagnosis: OL/OH = 1.20–1.30, OL/SuL = 1.55–1.80. Otoliths oval with thick profile. Dorsal rim dome-shaped, evidently elevated anterior to midline; ventral rim smoothly curved. Sulcus moderately wide, poorly differentiated into ostium and cauda. Cauda short with an obtuse posterior tip.

Description: Otoliths oval to elliptic, thick; inner and outer faces highly convex. Anterior and posterior rims pointed in holotype, blunt to nearly vertical in juvenile paratype. Dorsal rim dome-shaped, elevation just anterior to midline. Ventral rim smoothly curved. Sulcus median, very slightly inclined (∼15°), with ostium and cauda only faintly differentiated due to the indistinct collum, resulting in a nearly continuous sulcus. Ostial channel nearly indiscernible in holotype, but narrow, vertical, and ending just before dorsal elevation in paratype. Cauda short, extending slightly posterior to dorsal elevation; bears obtuse, truncated posterior end.

Remarks: Chiloconger is distinguished from other congrids by the notably short cauda, a character state considered plesiomorphic (Schwarzhans, 2019; Schwarzhans & Nielsen, 2021). Although based on small and not fully morphologically mature specimens, the new species differs from the two extant species, C. dentatus (Garman, 1899) and C. philippinensis Smith & Karmovskaya, 2003, by having a less pronounced, more anteriorly positioned dorsal elevation (Schwarzhans, 2019). Additionally, compared to the Early Miocene Chiloconger chilensis Schwarzhans & Nielsen, 2021 from Chile (Schwarzhans & Nielsen, 2021), C. aflorens exhibits a more compact and rounded shape.

Occurrence: Currently known only from the Piña Norte locality, Panama (Upper Miocene, Chagres Formation).

Genus Rhynchoconger Jordan & Hubbs, 1925	
Rhynchoconger sp.	
(Fig. 3C)	

Remarks: A single, fragmentary otolith exhibiting key Rhynchoconger characteristics is identified to the genus level. The preserved portion shows a discernible ostial channel and anterior ostium outline, diagnostic for Rhynchoconger (Schwarzhans, 2019). Due to the incomplete preservation, further species-level identification is not possible.

Order Argentiniformes	
Family Argentinidae	
Genus Argentina Linnaeus, 1758	
Argentina sp.	
(Fig. 3D)	

Remarks: A single thin otolith is assigned to the genus Argentina based on a strong, nearly orthogonal posterodorsal angle, a straight posterior rim, a curved ventral rim, and a median, horizontally oriented sulcus. The anterior portion of the specimen is missing, preventing confident assignment to species level.

Order Stomiiformes	
Family Sternoptychidae	
Genus Polyipnus Günther, 1887	
Polyipnus sp.	
(Figs. 4A–4B)	

Remarks: Polyipnus otoliths are distinctive by their tall, compressed shape, indistinctive sulcus outline, thin, slender rostrum bearing most of the ostium, elevated colliculum crest along the crista inferior, and a considerable thickness in the posterior rim. However, species-level identification is challenging due to extensive interspecific overlap and the limited availability of modern otolith reference material from the region, and the rostrum is very fragile and usually broken off in the fossil record, seriously hampering species definitions and recognition. Additionally, most specimens from the Chagres Formation are poorly preserved, with only the thick posterior rims preserved. The better-preserved specimens are depicted here.

Figure 4 Otoliths of Sternoptychidae and Myctophidae from the Upper Miocene Chagres Formation, Caribbean Panama.

(A and B) Polyipnus sp., ASIZF 0100947–0948. (C) Benthosema pluridens Schwarzhans & Aguliera, 2013, ASIZF 0100949. (D) Bolinichthys sp., ASIZF 0100950. Images are inner views. Scale bar = one mm.

Order Myctophiformes	
Family Myctophidae	

Remarks: The abundance of myctophid otoliths in the Chagres Panama is extraordinary. Most identifications are based on large subadult to adult individuals; tentative assignments for juvenile or poorly preserved specimens were conservative, resulting in a significant number of otoliths classified as Myctophidae indet. Consequently, the true myctophid diversity is likely underestimated in this collection. Significant advances in myctophid otolith taxonomy, sourced by the development of comprehensive global reference collections (Rivaton & Bourret, 1999; Schwarzhans, 2013b), have greatly improved identification in fossil assemblages (Schwarzhans et al., 2022; Lin et al., 2023b). We primarily follow Schwarzhans & Aguilera (2013) for taxonomic treatment and species-level identification of the myctophid otoliths in this study.

Genus Benthosema Goode & Bean, 1896	
Benthosema pluridens Schwarzhans & Aguilera, 2013	
(Fig. 4C)	
2013 Benthosema pluridens; Schwarzhans & Aguilera: pl. 1, figs. 8–12.	

Remarks: Otoliths of B. pluridens are relatively common in the collection. They are characterized by a sub-rectangular outline, a relatively flat dorsal rim, and multiple ventral denticles. However, juvenile myctophid otoliths often display intermediate outlines between rounded and sub-rectangular forms. To ensure accuracy, only specimens closely matching those illustrated by Schwarzhans & Aguilera (2013: pl. 1, figs. 8–12) were assigned to B. pluridens; other, less definitive specimens were conservatively classified under Myctophidae indet. Therefore, the true abundance of B. pluridens may be underestimated.

Genus Bolinichthys Paxton, 1972	
Bolinichthys sp.	
(Fig. 4D)	

Remarks: A single, juvenile otolith is assigned to Bolinichthys based on its prominent rostrum and gently curved, oblique posterior rim, consistent with diagnostic features of the genus (Rivaton & Bourret, 1999). Notably, Bolinichthys otoliths have not been previously reported from the Neogene of tropical America (Schwarzhans & Aguilera, 2013).

Genus Dasyscopelus Günther, 1864	

Remarks: Following the molecular phylogenetic revision by Martin et al. (2018), seven species previously assigned to Myctophum (M. asperum, M. brachygnathos, M. lychnobium, M. obtusirostre, M. orientale, M. selenops, and M. spinosum) were reallocated to Dasyscopelus. We adopt this updated classification herein, although we note that, in our view, otolith morphology among these species does not consistently show clear differentiation from other Myctophum species or internally within Dasyscopelus itself (see Schwarzhans & Aguilera, 2013: pl. 3). Otoliths attributed to Dasyscopelus are typically characterized by a pronounced posterior extension and a pointed ventral rim, giving them a more angular appearance compared to the generally rounded and often deeper-bodied otoliths of Myctophum. Exceptions include D. brachygnathos and D. selenops, which possess a much shorter posterior rim and a taller overall shape (Schwarzhans & Aguilera, 2013; Ng et al., 2024a).

Three closely related species—Dasyscopelus degraciai (Schwarzhans & Aguilera, 2013), Dasyscopelus jacksoni (Aguilera & Rodrigues de Aguilera, 2001), and the newly described Dasyscopelus inopinatus sp. nov. (see below)—are here allocated to the genus Dasyscopelus due to their otoliths having a protruding posterior part, a relatively flat dorsal rim, and general similarity to extant Dasyscopelus species, such as D. asper and D. lychnobius.

Dasyscopelus degraciai (Schwarzhans & Aguilera, 2013)	
(Figs. 5A–5F)	
2013 Myctophum degraciai; Schwarzhans & Aguilera: pl. 5, figs. 1–5.	

Remarks: The otoliths of D. degraciai are distinguished by their compact outline, less pronounced posterior extension, gently curved dorsal rim, and narrower sulcus. They differ from the co-occurring D. inopinatus sp. nov. and the Pliocene D. jacksoni by their shorter posterior tip, more compact, less angular shape, narrower ostium, and the more delicate crenulation of the otolith rims.

Dasyscopelus inopinatus sp. nov.	
(Figs. 5G–5L)	

Holotype: ASIZF 0100962 (Fig. 5I), Piña Norte, Panama. Upper Miocene, Chagres Formation. OL = 3.61 mm, OH = 2.43 mm.

Figure 5 Otoliths of Dasyscopelus (Myctophidae) from the Upper Miocene Chagres Formation, Caribbean Panama.

(A–F) Dasyscopelus degraciai (Schwarzhans & Aguliera, 2013), ASIZF 0100951–0956. (G–L) Dasyscopelus inopinatus sp. nov., (G, H, J–L) paratypes, ASIZF 0100957–0961, (I) holotype, ASIZF 0100962. Images are inner views unless otherwise indicated. 1, ventral view; 2, inner view. Scale bar = one mm.

Paratypes: Five specimens: ASIZF 0100957–0961 (Figs. 5G–5H, 5J–5L), same data as holotype. OL = 3.28–4.06 mm, OH = 2.31–2.66 mm.

Additional material: 57 specimens, unfigured, same data as holotype.

Etymology: From Latin inopinatus (feminine inopinata) = unexpected, alludes to the surprising and remarkable discovery of this new species, which exhibits a mosaic of morphological features shared with closely related congeners.

Diagnosis: OL/OH = 1.40–1.65 (mean = 1.48, n = 10), OL/SuL = 1.15–1.25 (mean = 1.23, n = 6), OsL/CaL = 1.45–2.10 (mean = 1.76, n = 10). Elongate otoliths with thin profile. Dorsal rim flat, nearly horizontal, with slight or no posterior elevation and a postero-dorsal angle. Ventral rim curved, bearing around eight lobes or denticles. Posterior rim short, nearly vertically straight. Sulcus very wide, with large, rectangular ostium and squarish cauda.

Description: Otoliths elongate, thin; both inner and outer faces are relatively flat. Anterior rim bearing large, protruding rostrum and shorter but conspicuous antirostrum, separated by clearly defined notch (excisura), varies from shallow to deep. Dorsal rim nearly horizontally flat, occasionally slightly elevated posteriorly, forming postero-dorsal angle just anterior to marked postero-dorsal concavity. Posterior rim short and nearly vertically straight. Ventral rim broadly curved, bearing ∼eight lobes or denticles. Sulcus wide, median-positioned. Ostium large, deep, rectangular; cauda short, squarish.

Remarks: The otoliths of the new species exhibit an intermediate morphology between D. degraciai and D. jacksoni. They share with D. degraciai a compact, deep-bodied outline and a less elongate posterior part, but resemble D. jacksoni in having a horizontal dorsal rim, wide sulcus, and a postero-dorsal concavity. Notably, the original illustrations of D. jacksoni (as Lampadena jacksoni in Aguilera & Rodrigues de Aguilera, 2001: figs. 7.15–7.22) suggest a more elongated and posteriorly extended shape compared to the ones documented in Schwarzhans & Aguilera (2013: pl. 5, figs. 6–10), as also indicated by differences in OL/OH ratios (1.60–1.85 vs. 1.50–1.65), although this may reflect ontogenetic variation.

Schwarzhans & Aguilera (2013) proposed a species turnover from the Late Miocene D. degraciai to the Pliocene D. jacksoni at the boundary between Messinian and Zanclean and further suggested a linear relationship between the two species. However, the finding of D. inopinatus suggests a more complex evolutionary history, with this new species likely representing a transitional or closely related lineage to D. jacksoni.

Occurrence: Currently known only from the Piña Norte locality, Panama (Upper Miocene, Chagres Formation).

Genus Diaphus Eigenmann & Eigenmann, 1890	

Remarks: Diaphus represents the most abundant and diverse taxon in our collection. A total of seven species, D. aequalis, D. apalus, D. barrigonensis, D. dumerilii, D. multiserratus, D. pedemontanus, and D. rodriguezi, are recorded. Our identifications are primarily based on larger, more mature specimens, which exhibit sufficient diagnostic characters to allow confident assignment (see remarks under the family). The otolith taxonomy of Diaphus has been extensively revised by Schwarzhans & Aguilera (2013), whose detailed descriptions and high-quality images serve as the primary references for this study. Therefore, only brief remarks distinguishing among similar co-occurring species are presented here.

Diaphus aequalis Schwarzhans & Aguliera, 2013	
(Figs. 6A–6C)	
1992 Diaphus aff. D. brachycephalus Tåning, 1928; Nolf & Stringer: pl. 10, figs. 11–13.	
?1992 Diaphus sp. 1; Nolf & Stringer: pl. 10, fig. 19.	
1998 Diaphus brachycephalus Tåning, 1928; Stringer: pl. 2, fig. 2.	
2013 Diaphus aequalis; Schwarzhans & Aguilera: pl. 13, figs. 13–25.	

Remarks: The species is common but never abundant within the Piña assemblage. Its otoliths are morphologically most similar to smaller individuals of the co-occurring D. apalus and, to a lesser extent, D. barrigonensis (see below). It differs from D. apalus by its very rounded and compact outline, and by possessing only a subtle postero-dorsal concavity (vs. a pronounced, deeply indented concavity). Compared to D. barrigonensis, D. aequalis is distinguished by its shorter rostrum and gently curved dorsal and posterior rims (vs. steeply inclined dorsal rim and sharp posterior rim).

Diaphus apalus Schwarzhans & Aguliera, 2013	
(Figs. 6D–6F)	
2013 Diaphus apalus; Schwarzhans & Aguilera: pl. 13, figs. 1–10.	

Remarks: See remarks under D. aequalis.

Figure 6 Otoliths of Diaphus (Myctophidae) from the Upper Miocene Chagres Formation, Caribbean Panama.

(A–C) Diaphus aequalis Schwarzhans & Aguliera, 2013, ASIZF 0100963–0965. (D–F) Diaphus apalus Schwarzhans & Aguliera, 2013, ASIZF 0100966–0968. (G–K) Diaphus barrigonensis Schwarzhans & Aguliera, 2013, ASIZF 0100969–0973. (L–O) Diaphus dumerilii (Bleeker, 1856), ASIZF 0100974–0977. Images are inner views. Scale bar = one mm.

Diaphus barrigonensis Schwarzhans & Aguliera, 2013	
(Figs. 6G–6K)	
2001 Diaphus sp. 2; Aguilera & Rodrigues de Aguilera: figs. 7.7–7.8.	
2013 Diaphus barrigonensis; Schwarzhans & Aguilera: pl. 7, figs. 1–9.	

Remarks: See remarks under D. aequalis.

Diaphus dumerilii (Bleeker, 1856)	
(Figs. 6L–6O)	
?1976 Diaphus dumerilii (Bleeker, 1856); Nolf: pl. 3, figs. 8–14.	
1992 Diaphus sp. 1; Nolf & Stringer: pl. 10, figs. 18, 20–21, 23 (non figs. 19, 22).	
1998 Diaphus sp. 1; Stringer: pl. 2, fig. 3.	
2001 Diaphus dumerilii (Bleeker, 1856); Aguilera & Rodrigues de Aguilera: figs. 7.1–7.2.	
2013 Diaphus dumerilii (Bleeker, 1856); Schwarzhans & Aguilera: pl. 10, figs. 18–23.	

Remarks: The otoliths of D. dumerilii are most recognizable by their slightly elevated antero-dorsal rim and the narrow antero-ventral part of the rostrum. However, as noted by Schwarzhans & Aguilera (2013), these otoliths are morphologically inconspicuous and, in our view, confident identification is generally limited to larger, well-preserved specimens. Some otoliths assigned to D. dumerilii by Nolf (1976) from Neogene deposits in Trinidad display a more compact outline (e.g., pl. 3, figs. 8, 11–12), suggesting they may actually belong to different species. Nevertheless, distinguishing such variations based solely on the available figures remains difficult.

Figure 7 Otoliths of Diaphus and Diogenichthys (Myctophidae) from the Upper Miocene Chagres Formation, Caribbean Panama.

(A and B) Diaphus multiserratus Schwarzhans & Aguliera, 2013, ASIZF 0100978–0979. (C–F) Diaphus pedemontanus (Robba, 1970), ASIZF 0100980–0983. (G–J) Diaphus rodriguezi Schwarzhans & Aguliera, 2013, ASIZF 0100984–0987. (K) Diogenichthys sp., ASIZF 0100988. Images are inner views unless otherwise indicated. 1, ventral view; 2, inner view. Scale bar = one mm.

Diaphus multiserratus Schwarzhans & Aguliera, 2013	
(Figs. 7A–7B)	
2013 Diaphus multiserratus; Schwarzhans & Aguilera: pl. 12, figs. 4–11.	

Remarks: Otoliths of D. multiserratus are readily distinguished by their elongate outline, wide sulcus, and, most conspicuously, the presence of numerous minute denticles along the ventral rim. Although not a frequent species in the collection, it is typically represented by larger, well-preserved specimens.

Diaphus pedemontanus (Robba, 1970)	
Figs. 7C–7F	
1970 Porichthys pedemontanus; Robba: pl. 16, fig. 8.	
2013 Diaphus pedemontanus (Robba, 1970); Schwarzhans & Aguilera: pl. 9, figs. 1–4.	

Remarks: Diaphus pedemontanus closely resembles the co-occurring and most abundant species D. rodriguezi but can be distinguished by a high, more undulating dorsal rim and a short, nearly vertically straight posterior rim, whereas D. rodriguezi has a gently curved dorsal rim, a deeper and wider excisura, and a larger rostrum (see below). Otoliths of D. pedemontanus are widely recorded from the Miocene and Pliocene of the Mediterranean (Girone, Nolf & Cavallo, 2010; Lin, Girone & Nolf, 2015; Lin et al., 2017a) and have also been documented in the coeval Caribbean assemblages (Schwarzhans & Aguilera, 2013). Ontogenetic variation in D. pedemontanus is considerable (Brzobohatý & Nolf, 2000), although we note that Caribbean specimens tend to be smaller than their European counterparts.

Diaphus rodriguezi Schwarzhans & Aguliera, 2013	
(Figs. 7G–7J)	
2013 Diaphus rodriguezi; Schwarzhans & Aguilera: pl. 9, figs. 5–12.	

Remarks: See remarks under D. pedemontanus.

Genus Diogenichthys Bolin, 1939	
Diogenichthys sp.	
(Fig. 7K)	

Remarks: A single otolith, characterized by a rounded outline (OL/OH = 1.05) and a thick profile, is assigned to the genus Diogenichthys (see Schwarzhans & Aguilera, 2013). However, due to partial damage along the anterior rim and the absence of additional material for comparison, species-level identification was not attempted.

Genus Lepidophanes Fraser-Brunner, 1949	
Lepidophanes inflectus Schwarzhans & Aguliera, 2013	
(Fig. 8I)	
2001 Lampanyctus aff. latesulcatus Nolf & Stringer, 1992; Aguilera & Rodrigues de Aguilera: figs. 7.13–7.14. (note: the authorship of L. latesulcatus should be Nolf & Steurbaut, 1983).	
2013 Lepidophanes inflectus; Schwarzhans & Aguilera: pl. 6, figs 16–19.	

Remarks: Otoliths of this species are very small, but highly diagnostic within the assemblage. They are particularly recognized by a subtle ventral inflection along the ostial crista inferior, although this feature is variably preserved among specimens. These otoliths also resemble much to those of Lampanyctus latesulcatus from the Tortonian Mediterranean; however, L. latesulcatus displays a more compact outline and lacks the ventral inflection characteristic of L. inflectus (Nolf & Steurbaut, 1983).

Figure 8 Otoliths of Lepidophanes, Lobianchia, and Myctophum (Myctophidae) from the Upper Miocene Chagres Formation, Caribbean Panama.

(A–C) Lobianchia johnfitchi Schwarzhans & Aguliera, 2013, ASIZF 0100992–0994. (D–F) Myctophum arcanum Schwarzhans & Aguliera, 2013, ASIZF 0100995–0997. (G and H) Myctophum affine (Lütken, 1892), ASIZF 0100998–0999. (I) Lepidophanes inflectus Schwarzhans & Aguliera, 2013, ASIZF 0101000. Images are inner views unless otherwise indicated. 1, ventral view; 2, inner view. Scale bar = one mm.

Genus Lobianchia Gatti, 1904	
Lobianchia johnfitchi Schwarzhans & Aguliera, 2013	
(Figs. 8A–8C)	
2013 Lobianchia johnfitchi; Schwarzhans & Aguilera: pl. 14, figs. 10–15.	

Remarks: Lobianchia johnfitchi is readily distinguished from other myctophid otoliths by its wide sulcus, prominently elevated antero-dorsal rim, and strongly depressed postero-dorsal rim. A closely related extant congener, L. dofleini (Zugmayer, 1911), exists during the Miocene–Pliocene in the NE Atlantic and Mediterranean (Lin et al., 2017a). We follow Schwarzhans & Aguilera’s (2013) interpretation that L. johnfitchi persisted in the Caribbean until the Middle Pliocene, whereas L. dofleini continued its presence into the modern Atlantic and Mediterranean.

Genus Myctophum Rafinesque, 1810	
Myctophum affine (Lütken, 1892)	
(Figs. 8G–8H)	
1992 Myctophum sp.; Nolf & Stringer: pl. 10, figs. 14–15.	
2013 Myctophum affine (Lütken, 1892); Schwarzhans & Aguilera: pl. 4, figs. 9–12.	

Remarks: Myctophum affine is recognized by its very rounded and relatively flat otolith outline. The specimens from the Piña assemblage agree well with extant M. affine otoliths (Nolf & Stringer: pl. 10, figs. 16–17; Schwarzhans & Aguilera: pl. 3, figs. 17–18). It differs from the closely related fossil species Myctophum arcanum Schwarzhans & Aguliera, 2013 by having a shorter posterior extension and a slightly curved dorsal rim (see below).

Myctophum arcanum Schwarzhans & Aguliera, 2013	
(Figs. 8D–8F)	
2013 Myctophum arcanum; Schwarzhans & Aguilera: pl. 4, figs. 13–17.	

Remarks: See remarks under M. affine.

Order Gadiformes	
Family Macrouridae	
Genus Coelorinchus Giorna, 1809	
Coelorinchus sp.	
(Figs. 9A–9B)	

Remarks: Two incomplete otoliths are assigned to Coelorinchus based on the presence of the characteristic pince-nez-shaped sulcus (homosulcoid-type) and the presence of a collicular crest at the collum. However, due to the fragmentary preservation, further identification beyond the genus level is not possible.

Family Bregmacerotidae	
Genus Bregmaceros Tompson, 1840	
Bregmaceros sp.	
(Figs. 9C–9E)	

Remarks: Bregmaceros otoliths are representatives of the third most common family in the Piña assemblage, although nearly all specimens are poorly preserved. They typically show a heavily abraded inner surface, such that the ostial and caudal depressions are not preserved (Figs. 9C–9D), while the general outline remains intact, complicating detailed taxonomic assignment. The best-preserved specimen is illustrated in Fig. 9E. Due to their preservation state, the specimens are conservatively identified only to the genus level.

Order Trachichthyiformes	
Family Trachichthyidae	
Genus Hoplostethus Cuvier, 1829	
Hoplostethus boyae sp. nov.	
(Figs. 10A–10D)	

Holotype: ASIZF 0101006 (Fig. 10A), Piña Norte, Panama. Upper Miocene, Chagres Formation. OL = 5.54 mm, OH = 5.38 mm.

Figure 9 Otoliths of Macrouridae and Bregmacerotidae from the Upper Miocene Chagres Formation, Caribbean Panama.

(A and B) Coelorinchus sp., ASIZF 0101001–1002. (C–E) Bregmaceros sp., ASIZF 0101003–1005. Images are inner views. Scale bar = one mm.

Figure 10 Otoliths of Trachichthyidae, Carapidae, and Ophidiidae from the Upper Miocene Chagres Formation, Caribbean Panama.

(A–D) Hoplostethus boyae sp. nov., (A) holotype, ASIZF 0101006, (B–D) paratypes, ASIZF 0101007–1009. (E and F) Carapus sp., ASIZF 0101010–1011. (G) Lepophidium limulum Schwarzhans & Aguliera, 2013, ASIZF 0101012. Images are inner views unless otherwise indicated. 1, ventral view; 2, inner view. Scale bar = one mm.

Paratypes: Three specimens: ASIZF 0101007–1009 (Figs. 10B–10D), same data as holotype. OL = 2.60–6.24 mm, OH = 2.52–5.67 mm.

Etymology: Named in honor of Brígida De Gracia (Boya in Ngäbere, the language of the Ngäbe-Buglé people) for her outstanding contributions to scientific research, public communication, and outreach activities in Panamá. The Ngäbe and their ancestors have inhabited the Isthmus of Panama for millennia, developing traditional ecological knowledge deeply connected to marine productivity cycles. Historical records demonstrate the Ngäbe’s reliance on seasonal fish abundance driven by upwelling systems along Panama’s Caribbean coast (Cybulski et al., 2025), creating a meaningful temporal bridge between the ancient upwelling ecosystem preserved in the Chagres Formation and the traditional knowledge systems that have recognized and depended upon these productive marine environments through time.

Diagnosis: OL/OH = 1.00–1.15, OL/SuL = 1.15–1.25, OsL/CaL = 0.70–0.90. Tall, sole-shaped otoliths with thick profile. Dorsal rim dome-shaped, evidently elevated posterior to midline; ventral rim either horizontally straight or smoothly curved, occasionally with large undulations. Sulcus very broad, shallow, median, well-differentiated into ostium and cauda. Ostium subtriangular, opening widely antero-dorsally. Cauda broad, rectangular.

Description: Otoliths tall, sole-shaped, thick; outer face strongly convex, inner face nearly flat. Dorsal rim curved, markedly elevated posterior to midline. Anterior rim with obtuse, upward-directed rostrum. Posterior rim straight, strongly inclined between postero-dorsal and postero-ventral angles, especially pronounced in larger specimens. Ventral rim horizontally straight to smoothly curved, occasionally with large undulations. Sulcus shallow, very broad, median; ostium subtriangular, opening widely antero-dorsally with shallow colliculum; cauda rectangular, broad, shallow. Dorsal depression fan-shaped, moderately deep.

Remarks: Among the six extant Hoplostethus species inhabiting the pan-Caribbean and East Pacific (H. atlanticus, H. fragilis, H. mediterraneus, H. mento, H. occidentalis, and H. pacificus), H. boyae most closely resembles juvenile otoliths of H. occidentalis (Haimovici et al., 2024: p. 139, but see Conversani et al., 2017: pl. 8). Other extant species typically exhibit a more variable dorsal rim, including flattened or undulating forms, often with digitiform projections, which may be a reflection of ontogenetic variation (Kotlyar, 1996). The new species differs by its consistently gently curved, dome-shaped dorsal rim, observed across both juvenile and adult stages. Compared to fossil congeners, such as the European Miocene species Hoplostethus praemediterraneus Schubert, 1905, and the Pliocene Hoplostethus pisanus Koken, 1891, H. boyae exhibits a more elevated dorsal profile and a shorter posterior rim, making the overall outline more compact and erect.

Occurrence: Currently known only from the Piña Norte locality, Panama (Upper Miocene, Chagres Formation).

Order Ophidiiformes	
Family Carapidae	
Genus Carapus Rafinesque, 1810	
Carapus sp.	
(Figs. 10E–10F)	

Remarks: All Carapus otoliths in the Piña assemblage are small and represented by juvenile specimens. They closely resemble an undescribed fossil Carapus otolith illustrated by Schwarzhans & Aguilera (2016: fig. 12), although preservation quality in our material is poorer. Given the incomplete preservation and small size, the specimens are conservatively assigned to the genus level. Schwarzhans & Aguilera (2016) further referred to a larger specimen from the Pliocene of Jamaica, depicted by Stringer (1998), but in our view, confirming such a connection requires additional material.

Family Ophidiidae	
Genus Lepophidium Gill, 1895	
Lepophidium limulum Schwarzhans & Aguliera, 2016	
(Fig. 10G)	

Remarks: A comprehensive review on the otolith taxonomy of fossil Ophidiidae from the region, and particularly the genus Lepophidium, has been provided by Schwarzhans & Aguilera (2016). The juvenile otoliths assigned to Lepophidium limulum closely match the type material illustrated by Schwarzhans & Aguilera (2016: fig. 56). The species is characterized by a gently declining dorsal rim, a moderately proportioned sulcus with a slight ventral notch at the posterior of cauda.

Order Gobiiformes	
Family Opistognathidae	
Genus Opistognathus Cuvier, 1816	
Opistognathus sp.	
(Fig. 11A)	

Remarks: A peculiar, thickset otolith is assigned to Opistognathus based on its distinctive sulcus morphology. The ostium curves sharply upward anteriorly, while the cauda initially bends slightly upward before flexing ventrally in its posterior portion. This sulcus configuration matches the pattern seen in extant Opistognathus otoliths (see Nolf & Stringer, 1992: pl. 15, fig. 10). Due to limited material, identification is restricted to the genus level.

Order Carangiformes	
Family Carangidae	
Carangidae indet.	
(Fig. 11B)	

Remarks: Two thin otoliths are assigned to the family Carangidae based on their pronounced concavity of the outer face and the presence of a typical percomorph-type sulcus. However, due to incomplete preservation, particularly of the anterior regions, further identification to genus or species level was not attempted.

Order Acropomatiformes	
Family Malakichthyidae	
Genus Malakichthys Döderlein, 1883	
Malakichthys schwarzhansi sp. nov.	
(Figs. 12A–12F)	
?1999 Epigonus denticulatus Dieuzeide, 1950; Aguilera & Rodrigues de Aguilera: pl. 1.	

Holotype: ASIZF 0101015 (Fig. 12A), Piña Norte, Panama. Upper Miocene, Chagres Formation. OL = 2.78 mm, OH = 2.29 mm.

Paratypes: Five specimens: ASIZF 0101016–1020 (Figs. 12B–12F), same data as holotype. OL = 1.89–4.55 mm, OH = 1.53–4.14 mm.

Additional material: Nine specimens, unfigured, same data as holotype.

Etymology: Named in honor of Werner Schwarzhans (Natural History Museum of Denmark) for his outstanding contributions to the study of fossil and extant fish otoliths, particularly in tropical America.

Diagnosis: OL/OH = 1.10–1.30 (mean = 1.20, n = 6), OL/SuL = 1.05–1.15 (mean = 1.10, n = 4), OsL/CaL = 0.60–0.90 (mean = 0.61, n = 6). Pentagonal otoliths with thick profile. Dorsal rim gently angled, highest anterior to midline; ventral rim gently angled or curved, deepest slightly anterior to midline; posterior rim nearly vertically straight. Sulcus broad, median, shallow, clearly divided into ostium and cauda. Ostium oblong, filled with colliculum, opening widely antero-dorsally. Cauda horizontally straight, narrow, slightly flexed at tip, nearly reaching posterior rim.

Description: Otoliths pentagonal, thick; thickness mostly from convex outer face umbo, inner face slightly convex. Dorsal rim gently angled, highest point anterior to midline, ending in postero-dorsal angle (most manifest in larger specimens). Ventral rim curved or subtly angled, deepest point slightly anterior to midline. Posterior rim nearly vertically straight. Sulcus broad, median, bounded by well-developed cristae. Ostium oblong, filled with colliculum, opening widely antero-dorsally; ostial crista superior markedly bent antero-dorsally, crista inferior gently curving upward. Cauda horizontally straight, narrow, nearly reaching posterior rim, slightly flexed at tip. Dorsal depression shallow, wide.

Remarks: The pentagonal outline of M. schwarzhansi is superficially similar to otoliths of several other families, such as Lactariidae and Epigonidae. However, none of these possess the markedly upward-directed ostium and sharply bent ostial crista superior observed in Malakichthys (Lin et al., 2023b; Ng et al., 2024b) (Fig. 13). Small specimens of M. schwarzhansi also resemble otoliths of Ambassis (Ambassidae), but Ambassis otoliths differ by having a more pointed posterior rim and a slightly widened caudal tip, features not observed in Malakichthys (see Fig. 14).

Figure 11 Otoliths of Opistognathidae and Carangidae from the Upper Miocene Chagres Formation, Caribbean Panama.

(A) Opistognathus sp., ASIZF 0101013. (B) Carangidae indet., ASIZF 0101014. 1, ventral view; 2, inner view. Scale bar = one mm.

Figure 12 Otoliths of Malakichthys (Malakichthyidae) from the Upper Miocene Chagres Formation, Caribbean Panama.

(A–F) Malakichthys schwarzhansi sp. nov., (A) holotype, ASIZF 0101015, (B–F) paratypes, ASIZF 0101016–1020. Images are inner views unless otherwise indicated. 1, ventral view; 2, inner view. Scale bar = one mm.

Figure 13 Extant Malakichthys (Malakichthyidae) otoliths.

(A and B) Malakichthys wakiyae Jordan & Hubbs, 1925, (A) 86.1 mm SL, CHLOL 10514, (B) 61.4 mm SL, CHLOL 10493. (C and D) Malakichthys griseus Döderlein, 1883, (C) 93.7 mm SL, CHLOL 14678, (D) 102.5 mm SL, CHLOL 34344. (E–G) Malakichthys elegans Matsubara & Yamaguti, 1943, (E) 131.9 mm SL, CHLOL 8241, (F) 62.9 mm SL, CHLOL 8800, (G) 77.8 mm SL, CHLOL 2605. (H and I) Malakichthys formosus Ng, Liu & Joung, 2023, (H) 72.1 mm SL, CHLOL 27488, (I) 64.9 mm SL, CHLOL 31419. (J–L) Malakichthys barbatus Yamanoue & Yoseda, 2001, (J) 161.0 mm SL, CHLOL 27489, (K) 83.51 mm SL, CHLOL 35072, (L) 223.5 mm SL, CHLOL 33087. Images are inner views unless otherwise indicated. 1, ventral view; 2, inner view. Scale bar = one mm.

Figure 14 Extant Ambassis (Ambassidae) otoliths.

(A) Ambassis kopsii Bleeker, 1858, 69.8 mm SL, CHLOL 28447. (B and C) Ambassis miops Günther, 1872, (B) 22.7 mm SL, CHLOL 31077, (C) 22.3 mm SL, CHLOL 31078. (D and E) Ambassis urotaenia Bleeker 1852, (D) 46.8 mm SL, CHLOL 27116, (E) 34.1 mm SL, CHLOL 27115. (F and G) Ambassis interrupta Bleeker, 1853, (F) 34.6 mm SL, CHLOL 27119, (G) 47.8 mm SL, CHLOL 27120. Images are inner views unless otherwise indicated. 1, ventral view; 2, inner view. Scale bar = one mm.

A large otolith previously illustrated by Aguilera & Rodrigues de Aguilera (1999: pl. 1) may belong to this species, although it shows a more strongly elevated dorsal area, possibly reflecting ontogenetic variation. Additional specimens are needed to confirm this assignment.

The genus Malakichthys comprises eight extant species distributed in the Indo-Pacific (Yamanoue & Matsuura, 2004; Ng, Liu & Joung, 2023). Other members of the order Acropomatiformes, such as Parascombrops, display much wider geographic and stratigraphic distributions (Schwarzhans & Prokofiev, 2017). The occurrence of M. schwarzhansi in the Late Miocene of Panama suggests that the genus had a broader Neogene distribution than it does today.

Occurrence: Panama: Upper Miocene, Chagres Formation in Piña Norte locality, Colon. ?Venezuela: Lower Pliocene, Cubagua Formation, northwestern Venezuela.

Otolith density, sample coverage, and diversity indices

A total of 6,211 otoliths were collected from 34 bulk sediment samples, yielding an average otolith density of 278.80 ± 135.59 otoliths/kg (Table S1). Our otolith collection was represented by 31 taxa belonging to 12 families, plus nine additional specimens remaining indeterminate (Table 1; Fig. 15). Rank abundance of otolith families remains stable with or without the unweighted sample CH18-1-1 (Fig. 15). Sample coverage, based on specimen counts, reached 99.87%, indicating a high level of sampling completeness. Rarefaction curves based on species richness (0D) suggested that the estimated diversity could increase to approximately 35 taxa with additional sampling effort, with or without the unweighted sample (Fig. 16). However, rarefaction curves for Shannon (1D) and Simpson (2D) diversity indices approached asymptotes, indicating that the most abundant and dominant taxa were successfully captured (Fig. 16). This pattern suggests that any additional taxa would likely be rare and of low-abundance.

Figure 15 Rank abundance of otolith families in the Upper Miocene Chagres Formation, Caribbean Panama.

Assemblages are compared with (blue) and without (coral) unweighted sample CH18-1-1. Families are ranked by total abundance across all samples, and plotted on a log scale. Numbers within bars indicate total specimen counts (n). Binomial 95% confidence intervals (error lines) were calculated using Wilson’s method and represent uncertainty in abundance estimates relative to total sample size.

Figure 16 Rarefaction curves of otolith-based taxa (Hill numbers) represented by species richness (0D), Shannon diversity (1D), and Simpson diversity (2D).

Assemblages are compared with (left) and without (right) unweighted sample CH18-1-1. Shaded areas represent 95% confidence intervals based on 1000 bootstrap replicates.

Discussion

Taphonomy and preservation

Otoliths are exceptionally abundant at the Piña site and are readily visible on the surface of the exposed sediments. Closer examination reveals that the otoliths are not randomly or evenly distributed but instead exhibit distinct clustering patterns within the sediment layers (Figs. 2C–2D). This clustered distribution suggests that otolith burial was not continuous, but occurred episodically.

Piscivorous predation, digestion, and subsequent excretion are important processes in the formation of otolith assemblages in marine sediments (Schäfer, 1972; Nolf, 1985; Welton, 2015; Lin et al., 2019; Agiadi et al., 2022). Predator feeding events can result in the accumulation of thousands of otoliths, especially by large predators such as whales, dolphins, and tunas (Fitch & Brownell Jr, 1968; Lin et al., 2020). At Piña, fossils of marine predatory mammals (dolphins and predatory whales) as well as piscivorous billfishes and sharks are common (Fierstine, 1978; Vigil & Laurito, 2014; Carrillo-Briceño et al., 2015; Pyenson et al., 2015; De Gracia et al., 2022), supporting a predation-mediated accumulation model. Therefore, the clustered distribution of otoliths in the sediments is consistent with deposition from predator excretions rather than from background mortality or mass mortality events.

Moreover, otoliths appear closely associated with ichnofossils attributed to Ophiomorpha (Stiles et al., 2022; Figs. 2C–2D). This suggests that burrowing organisms may have contributed to the local redistribution and concentration of otoliths within their burrow systems, either by incorporating otoliths during their activities or perhaps by selectively concentrating organic-rich material containing otoliths (Fig. 2D). While this burrowing activity does not increase the overall abundance of otoliths in the sediment, it may create localized zones of higher otolith density within and around burrow structures.

Surface-exposed otoliths are, on the whole, heavily weathered, often cleaving in half and exposing their whitish internal structure. Better-preserved otoliths were obtained from excavated blue-grey sediments found around 1–10 cm deep into the exposed sediments, which is where we focused sampling. In cases where lower taxonomic assignment was not possible, this was usually due to specimens being juveniles rather than from poor preservation. Nonetheless, a substantial proportion of specimens are moderately eroded, resulting in loss of outline details, resulting in taphonomic scores of 2 or 3 following Agiadi et al. (2022). This, combined with the prevalence of juvenile specimens, contributed to a relatively high number of otoliths being assigned only to the family level (Table 1).

Paleoenvironmental and paleoecological implications

The otolith assemblage at Piña is extraordinary in both abundance and composition, providing compelling evidence for a unique paleoenvironmental setting. Otolith densities in the Chagres Sandstone at Piña are exceptionally high, with individual sediment samples frequently exceeding 300 otoliths/kg, and one sample exceeding 775 otoliths/kg (Table S1). For context, Stringer et al. (2020) reported that a clay interbed of the Oligocene Glendon Limestone in Mississippi, USA, yielded 811 otoliths/kg, which they suggest may reflect enrichment driven by piscivorous predators. Thus, the densities observed at Piña rank among the highest otolith densities ever recorded from fossil assemblages for which sediment weight was systematically measured (cf. Leonhard & Agiadi, 2023). This unprecedented abundance coincides with a unique taxonomic dominance, where mesopelagic lanternfishes (Myctophidae) constitute over 96% of otoliths and represent more than 50% of the taxa (18 out of 31). Other pelagic fishes, such as hatchetfishes (Polyipnus) and codlets (Bregmaceros), were also present, albeit at much lower frequencies (each approximately 1.5% of the total otolith counts). Shallow-water taxa are rare, represented only by Carapus and Opistognathus. Nevertheless, these three pelagic families (Myctophidae, Sternoptychidae, and Bregmacerotidae) were the top three most abundant families (Fig. 15) at the site, demonstrating the importance of mesopelagic fauna.

The depositional environment of the Chagres Formation has been debated, with interpretations ranging from bathyal depths based on benthic foraminifera and fish assemblages (Collins et al., 1999) to shallow nearshore settings based on ichnofossils and sedimentological evidence (Stiles et al., 2022). Body fossil assemblages from the Chagres Formation include several indicators traditionally associated with deeper marine environments. Similarly, the occurrence of taxa such as Coelorinchus, Hoplostethus, and Malakichthys in our otolith assemblage could be consistent with deeper depositional environments, as these demersal genera are primarily associated with bathyal depths in modern oceans (Schwarzhans, 2013a; Lin, Girone & Nolf, 2016; Lin et al., 2017b; Lin et al., 2018). The dominance of myctophids themselves, being oceanic mesopelagic fishes that undertake diel migrations between surface waters at night and deeper waters of 300 m or more during the day, has often been taken to support interpretations of deeper water deposition.

However, sedimentological and ichnofossil evidence presents a contrasting picture. Stiles et al. (2022) provide compelling trace fossil and sedimentological evidence for shallow-water deposition, identifying archetypal Cruziana ichnofacies assemblages and sedimentary structures indicative of storm wave base environments (<50 m depth). At the same time, the total absence of typical shallow-water indicators such as ariid lapilli in our collection, which are near-ubiquitous in many Neogene shallow-marine otolith assemblages in the Caribbean (Aguilera et al., 2020), strongly suggests a depositional setting distal from the coast. Furthermore, some elements of the ichnological and sedimentary record could also be consistent with energetic upper-bathyal gateway systems where persistent bottom currents (contourites) and intermittent downslope processes redistribute bioclasts. Miguez-Salas, Rodríguez-Tovar & De Weger (2021) demonstrated that similar trace fossil assemblages and contourites can develop in deeper marine environments within oceanic gateways (the Rifian corridor), where strong bottom currents create dynamic conditions even at bathyal depths. Such a mechanism may be certainly applicable to the Isthmus of Panama in the Late Miocene, where tidal differences between the Caribbean and Panama could have created a highly energetic system even at depth.

A critical spatial consideration is the current geographic position of the Chagres Formation outcrops. Located approximately 20 km from the modern shelf edge at modern sea level, a bathyal depositional setting (>300 m depth) would require exceptional tectonic displacement to account for the current coastal position while maintaining stratigraphic coherence. Even accounting for the active tectonics during isthmus formation (O’Dea et al., 2016), known processes cannot reasonably explain how bathyal deposits could be displaced >20 km landward over the ∼6–7 Ma timespan. Typical rates of crustal shortening and arc migration in active margins (1–5 mm/yr; Coates & Obando, 1996) would require implausibly high displacement rates to account for such repositioning from a beyond-shelf-edge setting to the current coastal position, irrespective of eustatic sea level changes. The depositional depth of the Chagres Formation clearly requires further investigation through integrated geotectonic, sedimentary, paleoecological, and relative and global eustatic sea level analyses. The extraordinary dominance of myctophids suggests environmental conditions that differed significantly from typical modern shelf or bathyal settings. While myctophids can occur in relatively shallow waters starting at about 50 m over continental shelves (Lin et al., 2019; Heard et al., 2021), they are typically rare at such depths (<20%) and represented by only a few species that tolerate shelf conditions (Schwarzhans, 2013a; Lin, Girone & Nolf, 2016; Lin et al., 2017b). However, the Chagres assemblage may represent a paleoenvironmental setting with no exact modern analogue. Stiles et al. (2022) proposed that the orientation of the Caribbean coastline during Chagres Formation deposition could have facilitated Ekman-style seasonal upwelling. During the Late Miocene, the coastline orientation may have been positioned such that trade winds flowing parallel to the coast generated northward transport of surface waters, creating favorable conditions for localized upwelling in the Chagres region. Such a configuration would be consistent with the extensive evidence for seasonal upwelling systems documented throughout the Caribbean during the Late Neogene (O’Dea et al., 2007; Grossman et al., 2019; Jackson & O’Dea, 2023).

Many Caribbean coastal systems experienced strong seasonal upwelling during the Late Neogene, as observed in isotopic fluctuations within fossil shells and intracolony variations in module size in bryozoans from Florida and the Dominican Republic to Costa Rica and the Isthmus of Panama (O’Dea et al., 2007; Grossman et al., 2019; Jones & Allmon, 1995; Anderson et al., 2017). Additionally, the ecological composition of other nearshore fossil assemblages around the Caribbean during this period strongly support high levels of upwelling and productive coastal ecosystems, many of which contain abundant otoliths from fishes indicative of high productivity (Allmon, 1992; Jackson & O’Dea, 2023) and taxa shared with the Chagres Formation (Aguilera & Rodrigues de Aguilera, 2001). These productive ecosystems subsequently collapsed at the end of the Pliocene, giving rise to the modern, aseasonal and oligotrophic Caribbean (Jackson & O’Dea, 2023).

To explore this question further, we analyzed the proportion of mesopelagic planktivorous (principally myctophids) otoliths relative to all other otoliths in 187 modern Caribbean shelf sediment dredge samples (data from O’Dea et al., 2007; Jackson & O’Dea, 2023). When plotted against depth (Fig. 17), these data reveal that modern myctophid otolith assemblages increase from nearshore environments, peaking at around 120–150 m, before declining towards the shelf edge at 200 m. While intriguing, however, this shelf-focused analysis is somewhat inconclusive as it does not include bathyal depth samples and therefore cannot address the full depth range where myctophids are most abundant in modern Caribbean shelfs.

Figure 17 Proportion of myctophid otoliths by depth in modern Caribbean dredge samples.

Mesopelagic planktivorous (principally myctophids) otoliths relative to all other otoliths in 187 modern Caribbean shelf sediment dredge samples are plotted (data source: O’Dea et al., 2007; Jackson & O’Dea, 2023).

Regardless of depth, the tropical upwelling system observed at Piña differs substantially from modern temperate upwelling systems like the colder Humboldt and California Currents, where anchovies, sardines, and other small epipelagic fishes typically dominate (Chavez et al., 2003)—taxa almost entirely missing from the Piña assemblages. Instead, we argue that the Piña ecosystem was shaped by three key factors: warm tropical temperatures, high productivity from seasonal upwelling, and intense predation pressure. High tropical temperatures would have increased metabolic rates, which when combined with elevated primary productivity from upwelling, would have created conditions where predation rates can be extremely high (cf. Kordas et al., 2022). This would, in turn, favor the selective survival of predator-avoiding planktivorous fishes like myctophids, whose diel vertical migrations serve as a principal mechanism of predator avoidance. Despite these adaptations, a substantial proportion of myctophids still fell prey to predators. However, the extraordinary productivity, coupled with rapid demographic turnover of these small fishes, would have sustained large prey biomass, resulting in the high abundances of myctophid otoliths preserved in the sedimentary record.

Evidence for high predation pressure at Piña is substantial, including numerous predatory shark taxa such as Otodus megalodon (Carrillo-Briceño et al., 2015), and the frequent occurrence of large predatory vertebrates (e.g., dolphins, billfishes) and abundant elasmobranch teeth at the Piña locality (Fierstine, 1978; Vigil & Laurito, 2014; Carrillo-Briceño et al., 2015; Pyenson et al., 2015; De Gracia et al., 2022). Particularly striking, but as yet unremarked, is the extraordinary abundance of cookiecutter shark teeth (Isistius), which while observed in other Neogene tropical American sediments, reach exceptional frequency at the Piña Chagres site (see Table S1). As Isistius is a facultative ectoparasite that often feeds on large marine mammals, fishes, and sharks (Papastamatiou et al., 2010), their abundance testifies to a remarkably high density of large-bodied animals. The Piña scenario parallels aspects of tropical upwelling systems in the modern Arabian Sea, where primary production predominantly channels relatively directly into mesopelagic fish communities (Gjøsaeter, 1984; García-Seoane et al., 2025). In these systems, high planktonic productivity supports dense aggregations of myctophids, bypassing some of the longer and more complex food chains, and this energy is efficiently transferred to higher trophic levels, particularly to apex predators.

The combination of warm temperatures, strong coastal upwelling, extremely high planktivore abundance, and intense predation pressure provides a plausible explanation for the ecological observations at Piña. The Piña assemblage therefore represents a remarkable fossil example of a Late Miocene upwelling-driven, mesopelagic fish-dominated ecosystem, providing valuable insights into trophic dynamics and ecosystem structure during a critical period in the formation of the Isthmus of Panama (Fig. 18). However, we explicitly acknowledge that depositional environment requires further work to resolve the contrasting evidence.

Figure 18 Reconstruction of Late Miocene mesopelagic fish-dominated ecosystem in Caribbean Panama.

The illustration highlights key taxa, including lanternfish, hatchetfish, billfish, Isistius sharks, Otodus megalodon, Isthminia panamensis, and Lepidochelys sea turtle. Artwork by Yun-Kae Kiang.

Conclusions

The exceptionally abundant fossil otolith assemblage from the Chagres Formation at Piña reveals an extraordinary dominance of mesopelagic myctophid fishes during the Late Miocene in Caribbean Panama. Our otolith collection, based on over 6,200 specimens, consists of 31 taxa belonging to 12 families, and the otolith densities are among the highest ever documented from fossil deposits. The Chagres assemblage is remarkable for the extraordinary dominance of the family Myctophidae, constituting over 96% of specimens. Taphonomic observations, including clustered otolith distributions and close associations with ichnofossils, indicate that otoliths entered the sediments mainly through predator–prey interactions with additional preservation facilitated by burrowing organisms. Our findings further reveal previously unrecognized ecological dynamics in ancient tropical coastal ecosystems, where mesopelagic fishes aggregated in response to nutrient-rich conditions, and intense predation efficiently transferred energy to apex predators. The Piña assemblage, therefore, represents a rare fossil record of a mesopelagic fish-dominated ecosystem linked to coastal upwelling during the Late Miocene (Fig. 18).

Supplemental Information

Supplemental Information 1 Densities of fish otoliths and Isistius teeth from the Upper Miocene Chagres Formation, Caribbean Panama

Supplemental Information 2 Raw data of the composition of otolith-based fish taxa of all samples from the Upper Miocene Chagres Formation, Caribbean Panama

We thank Brígida De Gracia for assistance with logistics and for facilitating access to relevant fossil collections. Fieldwork support was provided by Blanca Figuerola, Mila O’Dea, Lorenzo O’Dea, Jorge Morales, Katie Griswold, Ramiro J. Solís, Kimberly García-Méndez, Antoni Lombarte, Yehudi Rodriguez Arriatti, Javier Pardo, Laura Lardinois, Meng-Chen Ko, and Li-You Lin. We are also grateful to Hsin-Wei Liu (Biodiversity Research Center, Academia Sinica, BRCAS) for figure preparation and logistical support, Siao-Man Wu and Chieh-Hsuan Lee (BRCAS) for assistance with figures, Yun-Kae Kiang for the artwork, and Chi-Wei Chien for helpful initial discussions on ichnofossils. Finally, we thank editor Kenneth De Baets and reviewers Werner Schwarzhans, Konstantina Agiadi, and Gary Stringer for their constructive comments and reviews. We thank the Bytnar family for their long-standing support of paleoecological research on the Isthmus of Panama.

Additional Information and Declarations

Competing Interests

Author Contributions

Field Study Permissions

Data Availability

New Species Registration

The authors declare there are no competing interests.

Chien-Hsiang Lin conceived and designed the experiments, performed the experiments, analyzed the data, prepared figures and/or tables, authored or reviewed drafts of the article, and approved the final draft.

Aaron O’Dea conceived and designed the experiments, authored or reviewed drafts of the article, and approved the final draft.

The following information was supplied relating to field study approvals (i.e., approving body and any reference numbers):

Permits for collecting and exporting paleontological samples were issued by the Ministerio de Comercio e Industrias (MICI) in Panamá.

The following information was supplied regarding data availability:

The raw data are available in Table 1 and the Supplementary Files.

The following information was supplied regarding the registration of a newly described species:

Publication LSID:

urn:lsid:zoobank.org:pub:996A25D1-9CB7-4AAD-9041-0ABCF49710C5

Chiloconger aflorens sp. nov. LSID:

urn:lsid:zoobank.org:act:E2D2D0F4-02E8-4A9D-9C14-54CE914F6A4E

Dasyscopelus inopinatus sp. nov. LSID:

urn:lsid:zoobank.org:act:7EB95634-CDF3-4C69-BD02-D1D99BFC1074

Hoplostethus boyae sp. nov. LSID:

urn:lsid:zoobank.org:act:4776F85D-F286-4F6A-895E-EF3167D7CBC7

Malakichthys schwarzhansi sp. nov. LSID:

urn:lsid:zoobank.org:act:444C3112-70BA-4E3C-8C94-9B5847A919E3.

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
