# Peer review of "Remarkable dominance of myctophid otoliths in Upper Miocene Chagres Formation, Caribbean Panama"

_PeerJ, doi:10.7717/peerj.20155_

## Round 0.1 · original submission · Major Revisions

· Academic Editor

Major Revisions

You provide crucial abundant new data and analysis on otoliths from the Chagres Formation and an alternative hypothesis to explain their environmental/ depositional context, which I would love to see published. However, I feel there are some crucial points which need to be addressed before publication:

Highest otolith density: as noted by reviewer 3 you state in some instances that the otolith concentration or density (otoliths/kg of bulk sample) is the highest ever documented which is not entirely correct. Please rephrase those statement to one or among the highest ever documented (as you did already in the conclusion) and cite relevant references for the highest density (compare reviewer 3).

Stratigraphic terminology: Please use capitals when you are referring official Early/Middle/Late Miocene (compare reviewer 3)

Sampling and stratigraphy of samples: The arrow in Figure 2 suggest your samples derive from the same relative position in a sandstone unit which were laterally collected. Is this correct? The stratigraphic provenance of your samples should be more clearly described in text. Also, you mention lithostratigraphy in this figure but you do not list members or formations in Figure 2. Please explicitly mention the lithostratigraphic units (formations, members) for clarity (compare reviewer 3).

Missing references: reviewer 3 suggested additional highly relevant references on Carribean otoliths as well as otolith density and taphonomy.

Range of measurements of morphological traits: you provide measurements for new taxa but given you really provide a large sampling of various species, it would be appropriate to also provide a range of measurements of all species you found (compare reviewer 1). I also feel providing not just the extremes, but the average/median and sample size of measured specimens (compare reviewer 3) would be appropriate for at minimum the new species and the holotypes (compare reviewer 1)

Sensitivity analysis: Given the one sample is unweighted and contains the largest numbers of otoliths (compare reviewer 3), it would be crucial to show how removing or adding it alters the proportion of taxa in your sample. This could be done in multiple ways (showing the proportion of unweighted versus weighted samples in Fig. 15 and/or performing/comparing rarefaction curves including and excluding the unweighted sample. In addition, it would be good to discuss how sample preparation may have contributed to the preservation and assignment of taxa (compare reviewer 3).

Confidence intervals: To understand the significance and potential impact of sampling, it would be helpful to understand the binomial error bars or confidence intervals for the relative proportion of collected specimens attributable to each family listed in Fig. 15 compared to the total. Compare Raup (1991) and see for example De Baets et al. (2012; Fig. 5) or Takeda & Tanabe (2014; Fig. 9) for examples when using proportions. Alternatively or additionally, you could show the proportions of specimens attributable to families in the unweighted versus the weighted sample.

Upwelling interpretation: The upwelling interpretation is interesting but there may be other ways to explain your observations (compare reviewer 1). It is heavily based on a single study suggesting a shallow water setting while various other studies suggest an alternative interpretation. Also, other studies in similar context of gateways have alternative interpretations for the distribution of ichnofacies (e.g., Miguez-Salas et al. 2021). I feel more extensive and careful discussion on this hypothesis is needed (compare reviewer 1 and 3). Depending on strength of support for coastal upwelling, you may want to consider revising your title accordingly (e.g., remove coastal upwelling from your title or add a question mark). Just to be clear, I feel this hypothesis merits to discussed in detail, but it does not seem to be the only viable hypothesis (compare reviewer 1 and 3). Reviewer 3 also mentioned that Ariid lapilli are common in many Neogene Carribean shallow-marine otolith assemblages and if the lack thereof in your samples could also be consistent with the proposed coastal upwelling paleoenvironment. Also I feel the phrasing of lines 707-716 could be revised for clarity (switch for predator-avoiding to predator-driven feels confusing).

Please address these as well as all other points raised including those in annotated pdfs.

Although the reviewers suggested minor revisions, I feel the raised points are more substantial than minor revisions including some possible revisions in species assignments (e.g., Diaphus) and need for additional (sensitivity) analyses. I feel these points can be reasonably and feasible be addressed within a reasonable timeframe.

I look forward to receiving the revised manuscript.

Suggested references:

Bacon, C. D., Silvestro, D., Jaramillo, C., Smith, B. T., Chakrabarty, P., & Antonelli, A. (2015). Biological evidence supports an early and complex emergence of the Isthmus of Panama. Proceedings of the National Academy of Sciences, 112(19), 6110-6115.

De Baets, K., Klug, C., Korn, D., & Landman, N. H. (2012). Early evolutionary trends in ammonoid embryonic development. Evolution, 66(6), 1788-1806.

Miguez-Salas, O., Rodríguez-Tovar, F. J., & de Weger, W. (2021). The Late Miocene Rifian corridor as a natural laboratory to explore a case of ichnofacies distribution in ancient gateways. Scientific Reports, 11(1), 4198.

Raup, D. M. (1991). The future of analytical paleobiology. Short courses in paleontology, 4, 207-216.

Takeda, Y., & Tanabe, K. (2014). Low durophagous predation on Toarcian (Early Jurassic) ammonoids in the northwestern Panthalassa shelf basin. Acta Palaeontologica Polonica, 60(4), 781-794.


·

Basic reporting

This is a well written and interesting article that is well worth being published in peerj and that will trigger much interest in the ichthyology and palichthyology community.

The taxonomy is state of the art and the documentation to it is excellent. I found only one little instant of error (see comment to Fig. 7A). I recommend to add more information about sizes of the object, particularly the sizes of holotypes and the ranges of sizes in new species. Also there are mentionings of comparison of extant and fossil otoliths apparently of different sizes but without giving size details.

The conclusions are interesting and sound, particular to the explanation how this uniquely enriched otolith clusters may have entered sedimentation. The paleocology is mostly fine, but I doubt the shallow water setting. This is based entirely on sedimentary and trace fossil assemblages while all other fossils indicate deepwater origin. There are alternative explanations available for the trace fossil and sedimentological setting and I have made references to that (Miguez-Salas et al., 2021). At least I would srongly recommend some careful discussion about this interpretation.

I made a few further comments in the text but they are all minor in nature.

The authors are to be congratulated for their work and I am looking forward to see the article published soon.

Experimental design

no comment

Validity of the findings

Excellent study but see discussion on paleo-water depth as mentioned under 1. Basic Reporting and annotated in the pdf.

Additional comments

none

·

Basic reporting

The study "Remarkable dominance of myctophid otoliths indicates Caribbean coastal upwelling in late Miocene Panama" by Lin and O'Dea is clearly written and structured, and the references are all necessary and pertinent.

Experimental design

This study presents new findings on the Late Miocene fish fauna of the Caribbean Sea before it was disconnected from the Pacific Ocean, which are important both for our understanding of Miocene marine fish faunas, but also in terms of the provided reconstruction of the paleoceanographic regime in the area, which was the result of tropical climate and oceanic connectivity that is different from the present day. The Introduction provides a thorough overview of this paleoenvironmental setting and how this research fills in an important knowledge gap. The authors use standard methods of investigation and these are described appropriately.

Validity of the findings

Although, as the authors mention, the tropical marine fish fauna of the Chagres Formation has been part of previous broader studies, these failed to capture several new species that the authors now identified. In addition, the authors interpret the dominance of mesopelagic fishes in a shallow-water domain as a result of regional upwelling, which is something new. The results are robust and the conclusions well stated.

Additional comments

I have noted only some minor comments for the authors to address in the attached pdf.

·

Basic reporting

A. For the majority of the manuscript, the English is understandable and comprehensible. There are some places where terminology, punctuation, or word usage could be improved or revised to better match professional English. This also includes stratigraphic usage. Specific examples are provided below and denoted by line numbers:

Lines 1–3. Should be “Late Miocene” in the title as Neogene epochs have been ratified (see reference for line 17). Late is used because it is referring to geologic time. Also, strongly suggest including formational name in title for greater specificity.

Lines 17, 29, 42, 55, 67. On Line 17, it is referring to a rock unit, the Chagres Formation, and it should be Upper Miocene. Early and Late Neogene subseries and subepochs have been ratified, and the initial letter should be uppercase, i.e., Late Miocene (if referring to geologic time) or Upper Miocene (if referring to rocks or rock units). See Aubry et al. (2022) in Episodes. The use is inconsistent in the publication. For example, line 175 has “Late Miocene,” but it should be “Upper Miocene” since it refers to a rock unit. The same applies to Lines 298 and 593. Usage must be corrected and consistent. Line 29 should be “Late Miocene” (referring to time), and Line 42 is “Late Pliocene” since it refers to geologic time. Line 55 should be “Upper Miocene” since it refers to a formation (rock unit). Line 67 should be Late Miocene since it is referring to geologic time.

Line 28 and 699. According to Merriam-Webster, “nearshore” without hyphen is preferred in English.

Line 40. Comma not needed after “develop.”

Line 51. There is an extraneous “&” between “Aguilera & Rodrigues” that needs to be deleted.

Line 65. I am not sure if the use of a large bulk sample that was not weighed qualifies as “quantitative.” Part of the study is certainly quantitative with precise weights.

Line 81. Middle to Late Miocene since it is referring to geologic time (see line 17).

Line 86. The date given, 6.4 –5.8 Ma, for the Chagres Formation would place the formation in the Late Miocene (geologic time). This should certainly be denoted in the text when discussing the geological setting.

Lines 88 to 89. Suggest: “. . . and is primarily comprised of.” As written, it is not clear as to meaning.

Lines 90–91. If these are formal members, then it should be the Lower Toro Limestone Member, the Middle Rio Indio Siltstone Member, and the Upper Chagres Limestone Member. An alternative is to have the Lower Toro Limestone, the Middle Rio Indio Siltstone, and the Upper Chagres Limestone members. In this case, “members” would be lowercase since it is not part of the specific formal name.

Lines 197, 330, and 535. Specific name and location of type locality should be included for all new species rather than just “type locality.”

Line 250. Recommend stating the number of specimens, and this would quantify “relatively common.”

Line 316. A brief statement on any morphological features of the outer face would be helpful.

Line 405. Comma after D. rodriguezi does not appear necessary.

Line 455. Middle Pliocene (uppercase “M”).

Line 485. The term “pince-nez” is not commonly used in otolith morphology. Recommend that “homosulcoid-type” be used, and if not, add in parentheses after “pince-nez.”

Line 494. Bregmaceros otoliths are not a family as stated in the sentence. Recommend that it read, “Bregmaceros otoliths are representatives of the third most common family . . .”

Lines 731, 737, 750. Should be Late Miocene (uppercase “L”). Use “Late” as it refers to geologic time.

Line 742. Comma after “interactions” is not needed.

Line 748. Comma after “conditions” is not needed.

Lines 997, 1003, 1008, 1014, 1020, 1027, 1035, 1039, 1046, and 1050. As a rule, lower/middle/upper should be used when referring to rock units, i.e., formations. So, all of these should be Upper Miocene Chagres Formation (shows stratigraphic position).

B. Literature references appear to be mainly complete. However, there are some omissions that need to be inserted. These are given below according to line numbers. In the introduction, the paleontology of otoliths background (taxonomy, morphology, etc.) is not especially exhaustive, but this would vary with the expertise of the reader. It seems sufficient for most readers.

Lines 22–23. The statement, “The otolith density in the sediments is the richest known globally” is not accurate as a higher concentration or density of otoliths has been reported by Stringer, Starnes, Leard, and Puckett (2020). They noted a 1.17 kg sample from a clay interbed of the Oligocene (Rupelian) Glendon Limestone in Mississippi, USA, that yielded 811.1 otoliths/kg. See Line 680 comments for discussion and reference that needs to be added. The present statement in the manuscript must be modified.

Line 51. Study by Stringer (1998) in Jamaica needs to be added.

Stringer, G. (1998). Otolith-based fishes from the Bowden shell bed (Pliocene) of Jamaica: Systematics and Palaeoecology. Contributions to Tertiary and Quaternary Geology, Volume 35(1–4):147–160.

Also, a study by Stringer, Ebersole, and Ebersole (2020) included Neogene otoliths from Panama, Columbia, Ecuador, Trinidad, Venezuela, and Brazil.

Stringer, G., J. Ebersole, and S. Ebersole. 2020. First description of the fossil otolith-based sciaenid Equetulus silverdalensis n. comb., in the Gulf Coastal Plain, USA, with comments on the enigmatic distribution of the species. PaleoBios, 37.ucmp_paleobios_49670.

John E. Fitch also did a preliminary study of the otoliths of the Gatun Formation in Panama and was published in the Journal of Paleontology in 1984. This reference should be included

Fitch, J.E. 1984. Osteichthyan otoliths. In D.D. Gillette. A marine ichthyofauna from the Miocene of Panama and the Tertiary Caribbean Faunal Province. Journal of Vertebrate Paleontology 4(2):172–186.

Line 64. If the Stiles et al. (2022) paleoenvironment based on ichnofossils and sedimentation is accepted for the Chagres Formation, then the reasoning for accepting it rather than the numerous previous studies should be elucidated and justified. Several of the studies that are not accepted are quoted repeatedly in the text. This is especially important if the paleoenvironment is going to be interpreted as shallow marine with a coastal upwelling.

Line 651. Schafer (1972) has a thorough discussion of the death, disintegration, and burial of fishes, including otoliths. Ecology and Palaeoecology of Marine Environments, University of Chicago Press. This reference should be included.

A detailed discussion of the potential for fish species enrichment by otoliths is found in Welton (2015) and should be included and addressed.
“The Marine Fish Fauna of the Middle Pleistocene Port Orford Formation and Elk River Beds, Cape Blanco, Oregon”

Line 680. A higher concentration or density of otoliths has been reported. Stringer, Starnes, Leard, and Puckett (2020) reported that a 1.17 kg sample from a clay interbed of the Oligocene (Rupelian) Glendon Limestone in Mississippi, USA, yielded 811.1 otoliths/kg. Horizontally adjacent samples yielded lower concentrations. The extremely high concentration, after other considerations and possibilities, was postulated to be related to enrichment by Oligocene piscivorous predators, such as toothed whales and other marine mammals.

Stringer, G., J. Starnes, J. Leard, and M. Puckett. 2020. Taphonomic and Paleoecologic Considerations of a Phenomenal Abundance of Teleostean Otoliths in the Glendon Limestone (Oligocene, Rupelian), Brandon, Mississippi. Journal of the Mississippi Academy of Sciences 65(1):101.

C. The structure of the manuscript follows the format of professional article with a title, authors (affiliations), abstract, introduction, geological setting, materials and methods, results (including detailed systematics for new species), discussion (includes taphonomy, preservation, paleoenvironment, paleoecology, conclusions, acknowledgements, and references

There are 16 figures that appear necessary and relevant. Photographs of otoliths are clear with good resolution and clearly labeled. Some photographs do not show diagnostic morphologic features, but this is a result of poor preservation and not the photography. There is one table that is certainly important (list of taxa of the Upper Miocene Chagres Formation). The supplemental files clearly indicate the raw date for each of the 33 samples collected and analyzed by the researchers. A few comments regarding figures and tables and shown by line numbers are provided below.

Figure 2A. Is labeled as “Lithostratigraphy,” but the formation and members are not designated, and it is unclear as to what it represents. The stratigraphy should be included, clearly noted, and explained.

Table 1. Title should be Upper Miocene (uppercase “U”) since it refers to a rock unit, the Chagres Formation.

It would be very informative to have a total weight of the bulk samples as well as a total of the otolith specimens (labeled as “counts”) and Isistius teeth specimens on the table.

Need to explain why the average number of otoliths per kg has a ± number?

Explain why the Isistius teeth were listed on Table 1 and other sharks were not.

An extremely important question is the stratigraphic relationship of the 33 samples. Do they represent different vertical stratigraphic positions within the formation, or do they represent horizontal stratigraphic positions (i.e., all samples from the same stratigraphic level in the formation but apart from one another)? This needs to be clearly explained in the text and is very important in the interpretation of the samples.

Experimental design

2. Experimental design
The manuscript defines the objective of the research and how this is accomplished (Lines 65–69). The importance of the research is explained, and its implications for otolith paleontology in the Neogene of the Caribbean are provided.
For the most part, the methodology is explained so that the research could be duplicated if samples were collected. Researchers applied for the proper permit for collecting, which was approved. All figured specimens and other otoliths are stored at the Biodiversity Research Museum, Academia Sinica, Taiwan. The nomenclature for the new species appears to conform to the International Commission on Zoological Nomenclature and has been registered in ZooBank. A few comments on the methodology are given below with line numbers for identification.

Lines 111–112. If 33 samples weighting approximately 0.6 kg each were collected, then the total weight of the samples was approximately 19.8 kg. This is important information that should be stated. Another alternative is to give the exact weight of the 33 samples based on Table 1. It is very important that the stratigraphic relationship of the samples be detailed and explained. To me, this is essential to this research.

Why was a sample collected, but not weighed, and utilized? It seems that it should have been weighed as with the other samples. It can introduce bias into the study, especially with the weight unknown. The sample that was not weighed provided the largest number of specimens. This should be explained.

Lines 118–119. It should be explained why these techniques were utilized. It should also be addressed if the sodium sulfate solution could possibly affect the preservation of the aragonitic otoliths. As noted in the manuscript, an extremely large proportion of the otoliths were poorly preserved.

Line 174. Would be very helpful to have the total number of specimens for all new species, i.e., holotype, paratype(s), and other examined material. This is done on Line 301 for a new myctophid species as “Additional material,” but it is not done for new species on Line 174

Line 183. Since it is a new species, the convexity of the inner face and outer would be very helpful and diagnostic. A brief description of the outer face is also commonly included in otolith descriptions, especially for new species. This is highly recommended.

Validity of the findings

Validity of the Findings
The raw data for the otoliths, the systematic description of the otoliths, and the statistical analysis performed are all provided. Therefore, it appears that the findings should be valid,
Conclusions are well stated and are related to research’s objectives.

Line 722. Could there be other factors other than high predator density causing the abundance of Isistius? What is the abundance of other sharks in the samples?

Line 739. You state in the Abstract and in other places that the otolith concentration or density (otoliths/kg of bulk sample) is the highest ever documented. Now, in the conclusions, you state that the otolith density is among the highest. The later statement is correct, but the others stating unequivocally that it is the highest concentration globally are not and should be revised.

Additional comments

Overall, the manuscript is well-written, highly informative. The fossil otoliths are well described, and the taxonomy of the otoliths appears to be very accurate. It is an important contribution to the understanding of the Neogene otoliths of the Caribbean.

Line 122. It is to be presumed that there are no lapilli (utricular otoliths) in the large assemblage. A sentence as to why no lapilli occur in the Chagres Formation would be informative. Ariid lapilli are common in many Neogene Caribbean shallow-marine otolith assemblages as indicated by several references. Could this total lack of ariids be related to the proposed coastal upwelling paleoenvironment?

---

## Round 0.2 · Minor Revisions

· Academic Editor

Minor Revisions

Thank you for addressing the suggestions which make the article even easier to follow and of even higher relevance. I particularly appreciate the extra effort to perform extra analyses. I would love to see this comprehensive work published but there are two crucial points which remain to be addressed which mostly relate to the paleo-water-depth assessment (compare reviewer 1). I feel you already worked towards reconciling and discussing alternative viewpoints but there is some extra efforts needed to discuss and embrace/align all alternative interpretations. The main points are:

1) Interpretations of the trace fossils and contourites: You largely follow Stiles et al. (2022) which provides an interesting new interpretation for the depositional environment based on trace fossils and sedimentological criteria. However, these interpretations are consistent with other data provided for the site/region (see references and points summarized by reviewer 1). In this context, I agree with the reviewer that it would be crucial to mention previously suggested reference Miguez-Salas et al. (2021) and discuss possible alternative interpretations based on similar trace fossils and contourites in deeper environments in similar contexts which would bring indications of different types of data and their interpretation closer together

2) Mid-shelf versus Upper Bathyal setting: I greatly appreciate the discussion on relative dominance of myctophids in shelf environments. However, I agree with reviewer 1 that the discrepancy between less than 20% of myctophids in shelf environments versus 96% of myctophids in studied sediments of Chagres Formation needs additional discussion and explanation. In this context, the ecology of myctophids and their relative abundance in deeper bathyal settings (e.g., Schwarzhans 2013, Lin et al. 2017 – compare reviewer 1) should be also discussed in detail. I do feel (as opposed to the reviewer) that new figure 17 could still be useful in this discussion ideally by adding relative abundance of myctophids in deeper settings and clearly designating possible misleading interpretations and also highlighting the value (96%) of myctophids in your samples as point of reference. Obviously, we cannot entirely rule out that Miocene may have had conditions which are not entirely comparable to either modern setting so at least discussing the differences in plausibility between all interpretations and the designating the most parsimonious scenario(s)/interpretation(s) would be crucial.

Please make sure to address these points (which are succinctly and constructively summarized by reviewer 1) as well as other points raised by the reviewers.

I look forward to receiving the revised manuscript.


Suggested reference:

Miguez-Salas, O., Rodríguez-Tovar, F. J., & de Weger, W. (2021). The Late Miocene Rifian corridor as a natural laboratory to explore a case of ichnofacies distribution in ancient gateways. Scientific Reports, 11(1), 4198.

·

Basic reporting

The manuscript is much improved from its earlier version and I have rather few comments still to make, which are included in the attached word file and a response to the response letter.

There is, however, one particular aspect that I cannot agree with, and that is the explanation the authors present for their paleo-water-depth assessment. In my firm opinion it is not acceptable the way it currently stands and need revision. I have extensively commented in the response letter to the responses. At the very least, the discussion should include controversial information available and known.

Experimental design

no comment

Validity of the findings

Sound, except for paleo-water-depth assessment.

Additional comments

See my response at the end of my response letter

·

Basic reporting

I am very satisfied with all the changed made in the revised manuscript.

Experimental design

I am very satisfied with all the changed made in the revised manuscript.

Validity of the findings

I am very satisfied with all the changed made in the revised manuscript.

Additional comments

I am very satisfied with all the changed made in the revised manuscript.

---

## Round 0.3 · accepted · Accept

· Academic Editor

Accept

Your revisions address the concerns raised by the reviewer 1 and provide a more balanced approach concerning the paleodepth interpretation. The discussion is now even easier to follow and even broader relevance including raising awareness of the need of future studies to further resolve discrepancies between tectonic, sedimentary, paleoecological and sea level indicators in this unique system. I therefore see no reason to send it into review again. The only aspect I discovered concerns a small formatting issue concerning the reference list (Anderson et al. attached to previous reference on line 883 should start on the next line 884) which can be resolved during the proofing phase. I look forward to seeing this work published.